# What Are More Efficient Transportation Services in a Rural Area? A Case Study in Yangsan City, South Korea

**DOI:** 10.3390/ijerph191811263

**Published:** 2022-09-07

**Authors:** Chang-Gyun Roh, Jiyoon Kim

**Affiliations:** 1Innovation and Strategy Division, Korea Institute of Civil Engineering and Building Technology, Goyang 10223, Korea; 2Department of Highway & Transportation Research, Korea Institute of Civil Engineering and Building Technology, Goyang 10223, Korea

**Keywords:** city-type bus, demand-responsive transport, population decline, rural areas, Korea

## Abstract

Population aging and population decline are experienced not only in South Korea but also in other countries around the world. In particular, public transportation operations, which have been centered on existing large buses, are struggling with a continuous deficit owing to the rapid population decline in rural areas, thus leading to a social issue. To address this issue, nations worldwide have attempted to find various alternatives. In South Korea, voucher taxis and city-type buses have been newly supplied in rural areas as alternatives. In this study, six city-type bus routes implemented in Yangsan-si, South Korea have been intensively reviewed in particular. The planned routes and operation status of each bus route were compared and reviewed based on geographic information systems. Six improved demand-responsive transport (DRT) operation methods were studied based on the operation patterns of city-type buses that were operated differently from the planed routes. Through this, a more suitable DRT small bus operation model for each route was proposed. Our study results will be a foundational proposal for policy makers concerned with improving public transport services and supplying new services in rural areas.

## 1. Introduction

South Korea has faced a rapid demographic change owing to problems that include a rapidly aging society and low birth rates. Regional cities are entering a super-aged society due to low birth rates and the outflow of young population to metropolitan cities. This demographic change has been so rapid that it has coined a phrase called regional extinction, which has been recognized as a social issue [1]. This change is not limited to South Korea but has become a global phenomenon including in Europe, and it needs to be addressed. For example, Finland is concerned about the problem of older adults living in rural areas. The population in rural areas such as remote regions has decreased while the demographic proportion of older adults has increased, indicating the need to provide measures to address this problem [2]. In some severe cases, a regional city would be extinct if all of its residents die of aging. However, no country would neglect this situation.

Thus, nations are making efforts to prevent population decline and regional extinction in regional cities, particularly in rural areas, or at least attempt to delay that trend. South Korea has provided a basis for sustainable development of regional cities through the “Act on Balanced National Development” and the “Act on Innovation Cities” [1]. Japan, which is a neighboring developed nation to South Korea, is also implementing countermeasures against population decline. For example, Japan established the Headquarters for Overcoming Population Decline and Vitalizing Local Economy, which started the effort to prevent population decline due to low birth rates and an aging society earlier than South Korea [3].

There are largely two measures for older adults living in rural areas. One is to prevent aging in a region by relocating the older adults in the region to urban areas [4,5,6]; the other measure is to delay or prevent aging society by improving public services in the region [7,8,9,10,11]. The primary reason for the former measure of relocating older adults was related to the cost of supplying public services in that region [4,5,6].

Generally, rural areas lack social infrastructure. To overcome this, Romania improved accessibility to public transportation services through the expansion of public transportation services thereby avoiding the regional population extinction [12]. South Korea has made efforts to prevent the reduction in transportation services by combining the welfare concept with public transportation services. For this purpose, taxis and small buses are mainly used as public transportation means in South Korea. This study aims to discuss the public transport services intensively using a small bus including share riding. This study started from the following two questions: First, is welfare transportation based on small buses running in rural areas operating in the originally intended direction? Second, what should be changed for more sustainable operations? The conclusions obtained through this analysis are as follows.

## 2. Efforts to Improve Transportation Services in Rural Areas

### 2.1. Overseas Cases

Overseas, efforts have been made to address the problems in rural areas in connection with adjacent urban areas instead of drawing conclusions by limiting the area in the process of dealing with how transportation services in rural areas should be improved [13,14,15]. Such an approach comes from recognizing a rural area as a satellite town of the adjacent urban area, which absorbs the economic sphere of living of the rural area [16,17,18]. Industrial by-products produced in rural areas are transferred and sold in urban areas. Thus, transportation services in rural areas are supplied focusing on the trip to adjacent urban areas.

Accordingly, transportation services are also supplied together to support trips within rural areas. For internal passage in rural areas, demand-response public transportation (dial-a-ride) is typically used as a representative transportation service [19]. The U.S. Department of Transportation (U.S. DOT) suggests that transportation services including shuttle buses those travel fixed routes in rural areas can provide services required for the trips of older adults, people with disability, and those with low income living in the area. It further suggested that the main purpose of the trip using this limited transportation service was to receive medical benefits [20].

Public transportation services in rural areas support all trips of local residents including commute to work or school and for recreation, and their dependence is much higher than that of urban areas [19]. Nonetheless, the capacity of the supplied public transportation services is insufficient, resulting in limitations such as longer time to travel and low service operation frequency. This is because it is a process of providing transportation services that can be continuously improved and maintained as part of the Mobility Services for All Americans program [19] and accordingly, a relatively simpler form of services are provided compared with that of urban areas.

The reason for the proposal of dial-a-ride by the U.S. DOT as the transportation service in rural areas was that the demand for transportation services in rural areas was small and the demand distribution was random. If demand for regular transportation is sufficient, a regular fixed public transportation system is more efficient than a demand-responsive system. Moreover, even if demand is low but the demand distribution is constant, public transportation with a route timetable with a low frequency of operation may be more efficient. Based on this rationale, dial-a-ride is the most efficient public transportation service operation method in rural areas where the demand for public transportation is small and the distribution of use is random.

From this viewpoint, more recently, a method of using existing transportation means that includes car sharing in the region has emerged rather than supplying new vehicles supplying vehicles for the public transportation services by the government. To solve the lack of transportation services in rural areas, the Finnish government has introduced the concept of mobility as a service (MaaS) [21]. A case study that attempted the MaaS concept, which began in Europe, and extended the application of the MaaS to suburban regions was published. In particular, Eckhardt et al. (2018) reviewed a Finnish region as a case. A fund was supplied by the Ministry of Agriculture and Forestry of Finland to make efforts to improve public service in rural areas. To overcome the limitation of movement of people and goods (cargo) in rural areas, MaaS using the open digital platform was implemented [21].

However, the implementation of the MaaS platform in a rural area was met with unexpected difficulties. Generally, the MaaS platform is digital; the majority of users in rural areas were older adults—a demographic characteristic of rural areas—having difficulty in using the digital platform, new devices, and application installation. Additionally, the budget for the implementation of the new digital platform was substantial. From this viewpoint, when the MaaS platform was introduced in the rural area, the combination of additional optional services was recommended to secure the minimum cost required for the installation and operation [21,22].

The simplest method to solve the transportation problem in a rural area is to create a foundation for ride-sharing. This is to move residents who have the same or close destination together when local residents move using a private vehicle. The government can solve the problem only by providing a car stop mark for a joint ride along with policy support [23]. Note that this method may be inappropriate for the promotion and fostering policy considering issues such as crime that may occur in the process of moving with other users.

### 2.2. Cases in South Korea

In South Korea, demand-response public transportation (dial-a-ride) is used to supplement public transportation services in regions with insufficient public transportation. The transportation service supplied to insufficient transportation service regions is provided as the transportation welfare using the government fund. The legal basis [24] of such a welfare service has been stipulated and enforced as follows:

Article 4 (Passenger Transport Business Subject to License or Registration of the Relevant Mayor/Do Governor) (1) As per the provision of Article 4 (1), passenger transportation businesses subject to license, shall have a license from the relevant Mayor/Do governor shall be deemed the on-demand passenger transport business (hereinafter referred to as the “On-demand passenger transport business”).

(2) As per the provision of Article 4 (1), passenger transportation businesses subject to registration that shall be registered to the relevant Mayor/Do Governor shall be deemed the town bus transport business, chartered bus transport business, and special passenger transport business.

(3) As per Article 4 (3), “a passenger transport business prescribed by Presidential Decree” refers to the town bus transportation business.

Article 3 (Types of Passenger Transport Business) refers to the Passenger Transport Service Act (hereinafter referred to as the “Act”). As per Article 3 (2), route passenger transportation businesses and area-passenger transportation businesses as per the Paragraph (1) 1 and (2) in the same article are classified as follows:Route passenger transportation business
a.Intra-city bus transportation business.b.Rural and fishing bus transportation business: A passenger transportation business whose operation system is set in a single administrative district of mainly Gun (excluding Guns in metropolitan cities) using a vehicle prescribed by the Ordinance of the Ministry of Land, Infrastructure, and Transport. In such cases, its operation type is classified into intercity express bus, intra-city bus, and general bus as prescribed by the Ordinance of the Ministry of Land, Infrastructure, and Transport.

Article 39 (Route Operation Permit of Passenger Vehicles) (1) “On the grounds that it is necessary for areas without public means of transportation or on grounds specified in Presidential Decree in Article 82 (1)-2” refers to any of the following cases.

Cases of transporting customers in areas where public means of transportation (hereinafter referred to as “public means of transportation”) such as route buses and trains (including city trains; hereinafter the same shall apply) are not operated or the access is extremely inconvenient;Cases of temporarily transporting customers in regions where the public means of transportation cannot be operated;Cases of no public means of transportation in the place of the facility or the access are extremely inconvenient.

(2) In cases falling under Paragraph (1)-3, the operation section of passenger vehicles shall be deemed a section between the facility and the closest bus stop or train station from the facility.

On the above basis, the welfare transportation services that are currently operated are taxi and small bus services. A different type of service is provided with different vehicle sizes.

#### 2.2.1. Taxi Service: 9-Cent Taxi (Voucher Taxi)

The taxi service, called the “9-cent taxi” is a welfare transportation service in Korea. Its fare is very cheap (KRW 100–KRW 1000, compared with KRW 3000 up to 2 km in regional cities), targeting remote islands and mountainous regions. Its main purpose is to increase the convenience of the trip for local residents. Transportation businesses that run taxis operate in the form of compensating for transportation costs to pick up passengers and transportation to the passenger’s destination with taxes. This service is the minimum transportation welfare service that assists a passenger to move to an adjacent bus stop or downtown (village center) rather than transporting a passenger to the final destination. This taxi service was introduced in the New York Times [25].

Figure 1 (Figure 2 is the translation of the contents of Figure 1 into English, the same as the contents of Figure 1) shows the voucher for a 9-cent taxi. In this voucher, “village name”, “operation section”, “date of use”, and “user” are indicated. As shown in this figure, it is different from normal taxis, which are flexible to set a place of taking a taxi and stop, but the route is already set between two locations (town hall and administrative office). The welfare transportation service cannot be supplied unlimitedly considering budget constraints; hence, a limited number of vouchers are delivered to each resident every month. A person who is supplied with the voucher should fill their name in the user box when using the supplied voucher, and the taxi driver will be compensated later in the form of a settlement of expenses by the department in the local government in charge of welfare transportation service after submitting the received voucher from the user instead of the fare.

The welfare transportation service is additionally set up for each region and positioned using both central and local government budgets. Generally, for one person, 2–6 vouchers are supplied (different regionally) and the operation section is also different across regions. It is regarded as a highly convenient and essential transportation service for residents who live in regions where only walking is the sole means to travel, and even now, in 2022, the number of towns implementing this service continues to gradually expand [27].

#### 2.2.2. Service Using a Small Bus: City-Type Bus

While the taxi service supports short-distance trips in small towns, there is another service that uses a small bus (seating capacity of 24 passengers) or a full-size van (seating capacity of 11 or 15 passengers) for regions where a large bus (seating capacity of 45 passengers) with a fixed route was closed due to population decline. This service is called a “city-type bus” in Korea and is provided in some areas. In contrast with the name, it is not for city regions but is operated in a region where the public transportation service is insufficient. It is operated in existing downtown central districts and nearby towns in rural areas; because it was a region where previous route buses were operated that it was called a city bus. Its name is also part of the plan to be distinctive from the taxi service.

Taxi service has been extended and is operational in a large area in rural, remote islands, and mountainous regions in Korea whereas the city-type bus is a relatively new service that began operating in the last three years. It was first operated in Gyeongsangnam-do, located in the southeastern part of Korea. More specifically, it was first operated by two local governments: Gimhae-si and Yangsan-si in Gyeongsangnam-do. Yangsan-si is the first city to operate the service in 2020 and had six bus routes by 2021. As of 2022, eight routes are planned including two additional routes. Gimhae-si has relatively various public transportation facilities including a city train. Thus, it has only two routes with demand-responsive transport (DRT). Thus, this study targets Yangsan-si for a more detailed examination of the service.

Yangsan-si is located in southeast Korea and its area is 485.617 km^2^. The status of land use in Yangsan-si shows that 73.5% of the total land comprises forest fields, which is the largest, followed by rice paddies (5.8%), dry paddy fields (2.5%), orchards (0.2%), and pastures (0.8%). As such, more than 82.7% of the area is used for agricultural and livestock industries. Meanwhile, building and factory sites account for only 5.5%, an area similar to that of rice paddy fields. Thus, Yangsan-si is a typical rural area [28].

Figure 3 illustrates the location of Yangsan-si and the layout of the administrative districts in Yangsan-si.

According to the Yangsan-si [28], its population comprised 354,726 residents (males: 177,759, females: 176,967) at the end of 2021, which increased by 2497 (approximately 0.7%) compared with 352,229 at the end of 2020. The number of households was 153,732, and the average number of members per household was 2.3. Of the total population in Yangsan-si, 120,539 residents lived in Mulgeum-eup, whereas Wondong-myeon had the smallest population with 3362 residents.

Figure 4 presents the population distribution, and Figure 5 shows the proportion of the elderly population living in the region. The proportion of the elderly population was the lowest in Mulgeum-eup, whereas the northwestern region (Wondong-myeon), where the population was small, had a high aging rate. This proportional difference was due to the new town constructed in the Mulgeum area, near Busan Metropolitan City, because of to the concentration of the new influx population owing to the creation of jobs.

Figure 6 shows the city-type bus running on the No. 5 route in Yangsan-si. The city-type bus No. 5 targets the region where the public transportation service of the previous two fixed routes was operated. It is a new service route after closing the fixed route of public transportation service. The number of passengers in the previous two fixed routes was five to six a day (as of the route closing in 2020), which could not be maintained for the daily operation of a large bus (seating capacity of 45 passengers) with a fixed route. Because of this, two routes were closed and one small bus was dispatched. As a result, as of the first operation month (December 2020), 13 passengers a day used the service, which was indeed larger than that of the previous large bus operation.

Figure 7 shows the photo of the vehicle in service of the city-type bus No. 2 route. For a city-type bus, the number of passengers per day is approximately 10 and the service frequency and passengers are not many. Thus, it has a unique operation method in that one vehicle with a single driver runs each route. Generally, one service is provided every hour and if the booking is not made in a service round, the service is canceled for that round. It does not operate a separate call center and the passengers call the driver directly using the mobile phone number assigned for each vehicle. Once a passenger calls the number for each bus route, they can talk with the driver directly to book. For example, the 11-digit number written on the side of the vehicle in Figure 7 is the unique mobile phone number of the driver who runs the city-type bus No. 2 route. Once the passenger calls the number and books the bus stop for boarding, the bus is dispatched to the bus stop at the next service round based on the current time.

The distribution of the six routes that are serviced as of 2021 is shown in Figure 8. Table 1 presents the service method of the city-type buses. The shortest route is No. 3 (3 km one-way) and the longest route is No. 1 (16 km one-way). As such, the routes are not too long and circulate over a short distance. The service frequency is one per hour, and a rest time (approximately 15 min) must be provided for drivers even if every hour is scheduled to run a round owing to a booking call.

As presented in Table 1, six routes have different vehicle sizes and methods. They are operating with methods which are set considering the demand and trip methods of each region. As shown in Figure 8, Route 5 has an unconventional operation method where a single bus with the same route number is serviced in different regions during specific times (morning and evening commuting hours). In addition, because a single driver runs a route, service is not provided during lunch time (one hour) to secure the driver’s meal time. This is the measure that is put in place to provide the maximum service using the available budget in the region.

Table 2 presents each route’s service frequency and number of passengers which were acquired in cooperation with Yangsan-si along with Table 1. According to the data in Table 2, the city-type bus route No. 5 had the most passengers (up to 3314 passengers per month (100 per day)) and the city-type bus No. 1 route had the lowest with 43 passengers per month (one per day). These data were statistical data from December 2020, after the COVID-19 onset, to September 2021. It is expected that the 2021–2022 data were very low when trips were restricted due to the COVID-19 pandemic, as compared with Table 2 (five passengers per day in city-type bus No. 5 route as of August 2021).

## 3. Is the City-Type Bus Running in Korea Operating Well as Planned?

The city-type bus, serviced to provide customized routes for transportation in disadvantaged regions in Yangsan-si, is up and running with a budget of KRW 534 million as of 2022. It is all public budget, which is funded to provide the minimum transportation service for the trip of local residents. Because it is public transportation, each route is predetermined and operates with the DRT method. However, decision making is a dilemma—whether to run the service along the entire route even with only one passenger in one service round (one per hour). Because the service is run with DRT, it does not need to run in regions where there is no demand. From this viewpoint, the service route and actual service record for each bus route were compared to determine the actual operation pattern. Based on this, the more efficient operation method among them was determined.

### 3.1. Analysis of Operation Systems Using Digital Tachograph Data

To identify actual operation records, digital tachograph (DTG) data were acquired and analyzed. In South Korea, it is mandatory mounting and operating DTG in public transportation vehicles such as buses, taxis, and cargo vehicles are stipulated by law. Accordingly, taxi and bus operators are subject to calculate vehicle operation information (operation location and time, travel processing, travel time, start and end time, business hour and distance, etc.) and to submit it to the government (the Korea Transportation Safety Authority [KTSA]) periodically [31]. This research acquired and analyzed the DTG data of the six city-type bus routes in Yangsan-si for one month during September 2021 in cooperation with the KTSA. The DTG data information is presented in Table 3.

Table 4 presents actual DTG data, which were preprocessed to analyze the operation pattern using the quantum geographic information system (QGIS) ver.3.26.

### 3.2. Comparison between the Plan and Actual Operation Results

The planned route and driving record of the actual city-type bus were tracked on the map using QGIS. Through the difference between the planned route and the actual driving record, how real passengers and drivers operated the city-type buses were revealed. The actual driving records were derived by overlapping the whole driving records for one month. Thus, the darker the color is, the higher the service frequency is.

In case the bus operates in compliance with the planned route, the route map and operation result will appear exactly the same. The case where operation results and routes are drawn in different forms is an area that should be reviewed with interest in this study. This may be because the actual route is unsuitable for operation, or the user or driver has found a more efficient way. Considering this, the purpose of this study is to find a way to introduce a more efficient method. The planned route for each route and service status are as follows.

#### 3.2.1. City-Type Bus No. 1 Route

Figure 9 shows the city-type bus No. 1 route including the location of bus stops. Figure 10 shows the driving record of the actual bus. The analysis result shows that more than 80% of the city-type bus No. 1 route belonged to a mountainous district and the service frequency was low. Some of the driving records started from the ski resort, which indicated the route was also used for tourism purposes.

The number of passengers was not many, and the place lacked other public transport services, this service was operated continuously. In contrast with other routes, this route had no available detour, thus the bus drove accurately followed the planned route from the starting bus stop to the last.

#### 3.2.2. City-Type Bus No. 2 Route

The city-type bus No. 2 route is shown in Figure 11 and the actual driving record is shown in Figure 12. Originally, the route was planned in a Y shape, but only the route stretched to the north showed a solid thick line as shown in Figure 12, which verified that the trips were concentrated on a specific bus stop (located in the far north in Figure 11). Other bus stops did not have much demand; thus, the service did not run over the entire route but stopped in the middle of the service as shown in Figure 12.

#### 3.2.3. City-Type Bus No. 3 Route

The one-way length of the city-type bus No. 3 route is approximately 3 km and is relatively short. Although the route passes right beside of an industrial complex, the commuters there mainly used private cars. Thus, the use of the city-type bus No. 3 route did not lead to the industrial complex as is shown in Figure 13 and Figure 14. As the actual driving record in Figure 14 shows a homogeneous color regardless of specific sections or bus stops, the demand was distributed evenly throughout the entire route.

#### 3.2.4. City-Type Bus No. 4 Route

The city-type bus No. 4 route is a regular service operation during commuting hours (8:00–9:00 a.m. and 6:00–7:00 p.m.) and changes to a DRT operation at other times. As shown in Figure 15, the planned route was normally serviced for bus stops located in the upper right side (yellow line) and operated for the gray line section located on the left side during commuting hours. However, the actual driving verified the service was not provided even in the regular line service section and returned quickly without completing the route as indicated in the blue line in Figure 15. The section that had no planned route, indicated in the red ellipse in Figure 15, had a thick color as shown in Figure 16 even in the DRT service time. This indicated that the planned route was not matched by the actual demand or that the route was determined at the driver’s discretion for efficient travel.

#### 3.2.5. City-Type Bus No. 5 Route

Figure 17 and Figure 18 show the city-type bus No. 5 operation route during the normal time and actual driving record. The bus stop located in the far north is the depot, from which the bus moves in the south along the right line and then moves back to the north along the left line to return to the depot. The actual driving record in Figure 18 verified that the bus stop in the lower left end had a smaller demand than other bus stops, resulting in a low frequency of service record. Additionally, driving trajectories unmatched to planned routed were revealed which are in the middle of the route (section indicated by the blue ellipse in Figure 17). It means buses could come back to depot with faster and efficient course as the No. 5 Route was operated in full demand response method.

Figure 19 and Figure 20 show the operation of the city-type bus No. 5 route during commuting hours. The city-type bus No. 5 route adopted a unique operation method in which the general route in Figure 19 was not serviced for two hours (8:00–9:00 a.m. and 6:00–7:00 p.m.) but the route in Figure 20 was serviced (refer to Figure 8).

The commuting service of the city-type bus No. 5 route did not show any significant difference as is shown in Figure 17 and Figure 18. Note that unplanned operation in some sections (indicated by the blue line in Figure 16) was observed in the route depending on the demand location for the service.

#### 3.2.6. City-Type Bus No. 6 Route

The city-type bus No. 6 route is serviced to the industrial complex only during morning and evening commuting times (orange color route in Figure 21) and serviced in the purple color section in Figure 21 during the daytime.

Comparing Figure 21 and Figure 22, it is revealed that the shortest distance paths were chosen often and could be seen driving record through the path outside the route. In particular, the orange section in Figure 21 (the red circle in Figure 21), indicating the inside of the industrial complex, was different between the planned route and the actual driving record. This verified that the planned bus stops and route were different from the section and location demanded by passengers, which was indicated by driving to the unspecified location in the route or no driving record in the planned route.

The service route during the normal hours is as shown in the purple section in Figure 21 and was significantly different from the actual driving section, shown in Figure 22. The service frequency of the left end in the purple line was low as the demand low too, it resulted making a protruding trajectory on roads which are not planned to go (indicated by the blue circle in Figure 21). This was due to the discretion of the driver for the service only to travel to a place where the demand was requested.

## 4. The Most Efficient Public Transport Method in Rural Areas

The comparison results between the six routes of the DRT buses serviced in Yangsan-si, South Korea and their actual driving records verified two major patterns. One is that the buses stopped and did not go further places in the planned route as there was no demand and used the shortest path to complete a service round. Another pattern revealed is additional services to locations or sections that are not part of the service for a specific route. Additional services were provided owing to the bond between elderly passengers who uses the service daily and the drivers. For the convenience of elderly passengers, drivers served places that they are not permitted to stop at regardless of whether it is a violation. In this regard, it would be unreasonable to set a route map and require drivers to unilaterally follow the route. To consider this situation, this research recommends a methodology operating the DRT buses.

### 4.1. Operation Methods Using DRT Buses

Generally, it is convenient for DRT public transportation services to employ the semi-fixed operation method (Figure 23) as in the case in Yangsan-si that operates only upon service demand, which contrasts from the fixed method of operating with the traditional schedule and planned routes. The semi-fixed operation does not need high-level operation techniques and booking can be done through call centers or smartphone apps, responding only to the service requested.

If the degree of freedom of travel by DRT is extended further, the operation method can be divided into a case where the route is maintained and a case where it is not. If the route is maintained, the semi-flexible method—that can shorten or extend the service route depending on the degree of demand—can be applied. If the route is not maintained, the zoning method is more advantageous when the demand location is not consistent or varies often and the trip between adjacent regions occurs frequently. It is a method grouping regions where the trips frequently occur and setting the grouped regions to the DRT public transportation service zone thereby responding to the trip request individually within the zone.

For regions where there is a significant difference in demand for public transportation use over time, the concentrated demand response method can be applied. In the concentrated demand response method, the minimum trip service is provided by existing traditional public transportation methods that provide a fixed route service. It is a method of supporting a trip for passengers through demand response or adding more vehicles to the same route for excessive demand at specific times. This method is more suitable for regions where the supply is not sufficient compared with the demand such as in the outskirts of a city.

The transfer HUB is a method suitable for regions having a transit center which connects many other branch transportation services. Transferring passengers to the transit center using DRT rather than using fixed routes where public transportation demand is not sufficient. If a transit terminal is present between regions located in suburban and rural areas, operating multiple DRT services links to transportation terminal would be and appropriate method.

The above explanations are summarized as follows and shown in Figure 20.

Fixed: It is suitable for regions where regular demand is maintained and passengers are concentrated.

Semi-fixed: It is suitable for regions where passengers are concentrated but the demand is insufficient or significantly fluctuates over time.

Zoning: It is suitable for regions where trips within the regions adjacent are many and the start and end points are distributed (the most general form of DRT).

Semi-flexible: It is suitable for regions where regular demand and irregular demand are mixed.

Concentrated demand response: It is suitable for regions where the regular demand is maintained and exceeds the capable supply at peak time.

Transfer HUB: It can be applied where the transit center is located in adjacent but separated regions.

### 4.2. Discussion: Selection of Operation Method Considering Regional and User Characteristics

The public transportation services with regular routes are gradually closed toward the outskirts of Yangsan-si, South Korea due to the society’s aging and population decline. Thus, DRT means are used to maintain the trip service of local residents. To do this, a new DRT public service called the city-type bus was supplied. The city-type bus adopted the fixed operation method that is scheduled to service once per an hour and the routes and bus stops to be serviced are generally fixed, even though its operation method varies for each route. Nonetheless, it was not serviced at times without demand as it is operated with the DRT.

Due to the limitations of the operation method, the DTG data verified that routes that connected the start and end points using a shortest distance, which were not planned, were created and serviced during the process. Additionally, skipping route and stopping earlier than the planned route when there are no requests for use were visible patterns too. These are patterns violates the principle of service from the traditional route operation’s point of view. However, it was highly advantageous method both for drivers and passengers from the principle of operational efficiency and service (demand response).

If this type of service method is more suitable for local communities, researchers and policy makers should provide a method and basis that can allow its service under legal protection. This is because a more sustainable service basis can be provided in this way. Thus, it is a more desirable method for the current service method that is tacitly implemented to be systematized and combined with legally acceptable service technology.

According to the six methods in Figure 20, the methods used in the current city-type buses are close to the semi-flexible operation method. If the demand is low and the request locations (start and end points) are randomly distributed, a fully flexible method such as an on-demand taxi would be more suitable. However, as a public transportation service using a small bus is inefficient in terms of one-to-one boarding and deboarding service, a fully flexible method will degrade the service efficiency considering that passengers should be able to board and deboard continuously as a ride-sharing form.

Based on the six routes in Yangsan-si, a suitable DRT service pattern considering real service patterns and road circumstances is proposed. For the city-type bus No. 1 route, the semi-fixed operation method is most suitable because the route has no detours and specific bus stops are required frequently. The convenience of local residents will improve further if the route is serviced only upon demand and additional passenger boarding is allowed in the middle of the service. To do this, a technology that can book a bus in service in real time that is expected to arrive in the future will be needed instead of general public transportation boarding–deboarding way that simply takes passengers waiting at bus stops.

For the city-type bus No. 2 and 3 routes, which are characterized by routes servicing two divided regions, a modified semi-flexible method was applied. A modified method means that only one region where the demand is requested out of two regions is selectively serviced (only half of the route is serviced) in contrast with the semi-flexible method, which shortens a certain region during the service. The zoning method can be applied if one vehicle is serviced in each region. However, it cannot be applied if the demand in each region is low or the number of vehicles is insufficient. So, Yangsan-si could not adopt zoning method as well because vehicles are insufficient to operate. Rather than servicing two regions once per hour, only one region is serviced once per hour, and the other region is serviced later (for example, first round: Town A, second round: Town B, and third round: Town A).

For the city-type bus No. 4–6 routes, the semi-flexible method could provide more efficient services as particular bus stops were highly demanded and trip paths were prescheduled. The bus operation efficiency increases as it can passes courses and stops with no requests in its entire route. Using not-in-service vehicles owing to no demand or using the operation time saved through the shortened operation, additional services can be provided, thereby improving the service for users. Furthermore, if those times are used for rest time for drivers, additional service routes can be established through budget reduction.

Public transportation is a national service to provide a minimum mobility service fairly to all citizens. However, public transportation services should not be the same in all regions. If there are a public transportation service based on the demand from local residents already, it must be institutionalized and systematized.

## 5. Conclusions

As mentioned in the Introduction, discussion regarding urban and transportation issues has increased owing to global population decline and aging society. In particular, the efficiency of public transportation based on large buses, which provided support for large scale mobility previously, is rapidly decreasing especially in rural areas. Thus, governments experience substantial deficit in the process of spending budget to maintain the public transportation services. To address this issue, South Korea removed existing public transportation routes and adopted voucher taxis or on-demand bus models. Each model is selectively used considering the current regional situation, status, and residents’ characteristics. For taxi services, improvements of the current service are minimal in terms that taxis support the number of trips of public transportation to users limited to areas with small populations. Some additional alternatives are the extension of travel distance and additional supply of vouchers. Meanwhile, the city-type bus is a new service in South Korea adopted less than three years ago. It has been implemented around regions where existing public transportation routes were removed. It is a unique way of DRTs operated using one vehicle and one driver over a single route as a self-help measure. Thus, this study aimed to introduce this to policy makers who are concerned with such services and trying to find alternatives. It would be helpful to review this case for policy makers wanting to improve regional public transportation services and introduce city-type buses as a good alternative. Furthermore, additional improved alternatives are proposed by presenting the planned routes and actual operation status of six routes that are currently implemented in Yangsan-si, South Korea. This study’s results would not only help improve the current services in Yangsan-si but also provide various options to policy makers in worldwide who want to adopt such services.

DRT will be positioned as the main means of operation in the future to ensure the mobility right of local residents in rural areas where the demand for transportation is low and the request occurs irregularly in time and locations. The same or similar DRT method can be adopted not only in South Korea but also in similar regions around the world. In this process, our study results will be helpful in designing and determining services in relevant regions.

As a limitation of this study, it can be said that it has only proved how different the existing operation method is from the planned based on DTG data and showed how to implement the recommendation from the data for a more efficient method. In order to model and verify the method of granting additional degrees of freedom for routes, additional research including the setting of stops for individual routes must be accompanied. In the future, it is expected that comparative analysis of the operational efficiency of the existing route method and the new method and verification of the route setting methodology accompanied by formulas will be made.

## Figures and Tables

**Figure 1 ijerph-19-11263-f001:**
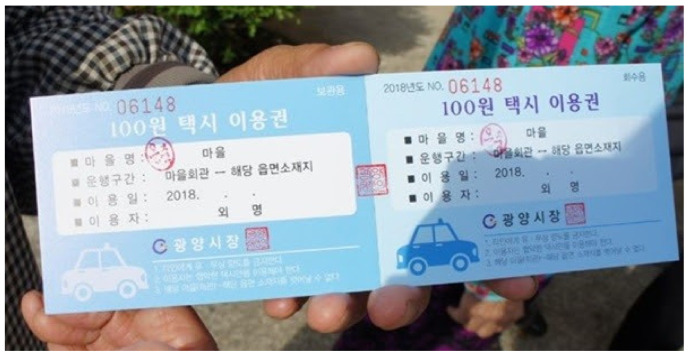
9-Cent taxi voucher [26].

**Figure 2 ijerph-19-11263-f002:**
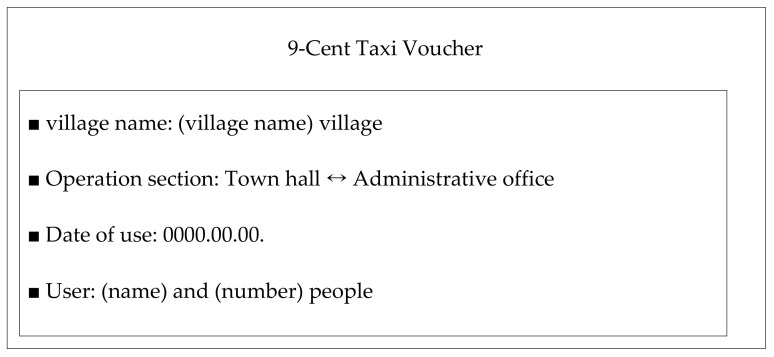
9-Cent taxi voucher contents in English [26].

**Figure 3 ijerph-19-11263-f003:**
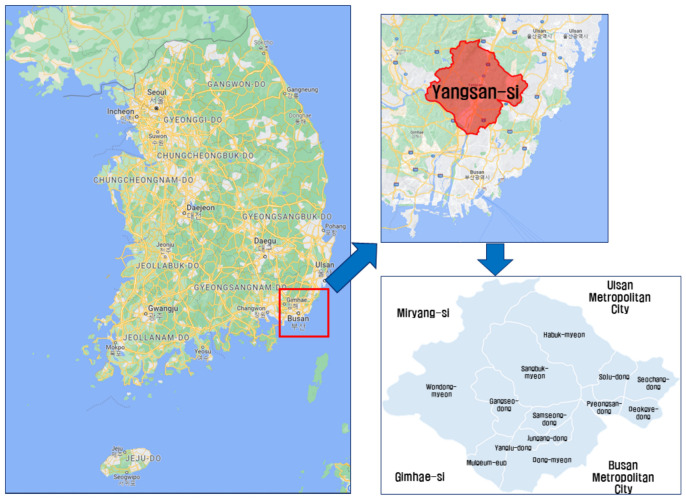
Location and administrative district diagrams of Yansang-si.

**Figure 4 ijerph-19-11263-f004:**
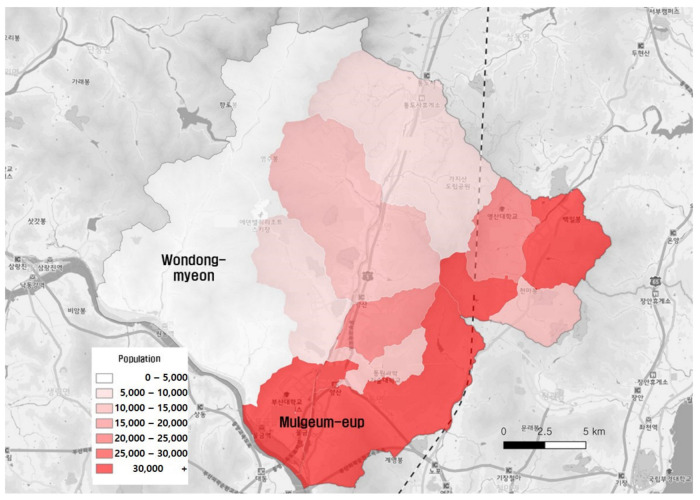
Population distribution in Yangsan-si.

**Figure 5 ijerph-19-11263-f005:**
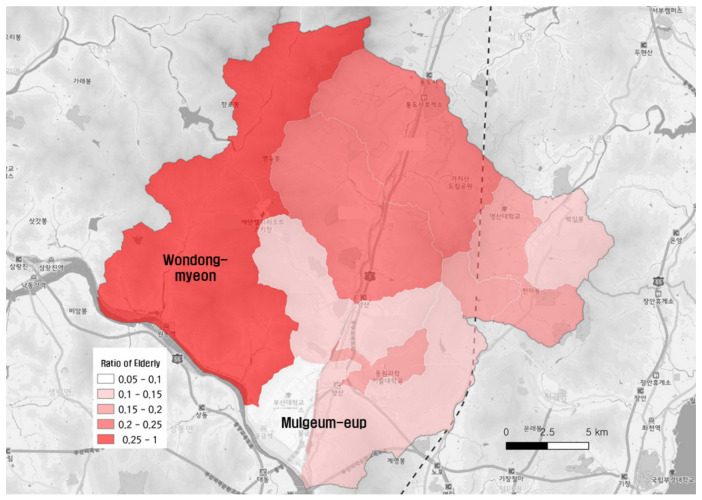
The proportion of the elderly population by administrative districts in Yangsan-si.

**Figure 6 ijerph-19-11263-f006:**
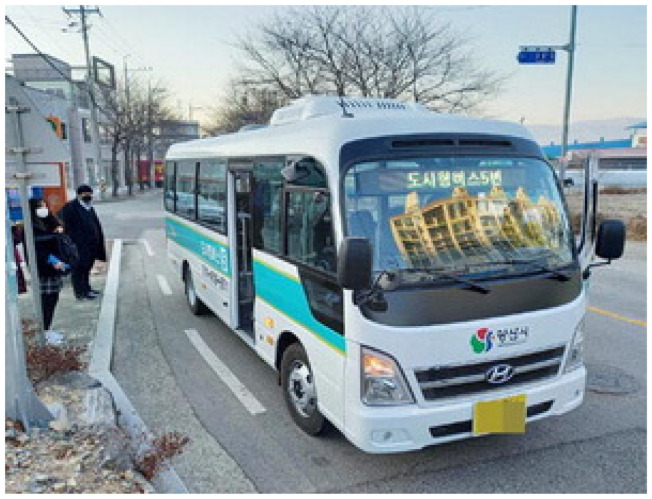
City-type bus in Korea (Yangsan city-type bus No. 5) [29].

**Figure 7 ijerph-19-11263-f007:**
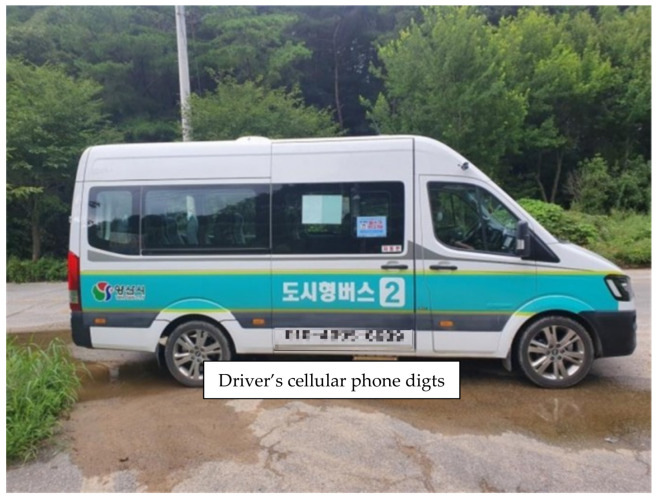
City-type bus in Korea (Yangsan city-type bus No. 2) [30].

**Figure 8 ijerph-19-11263-f008:**
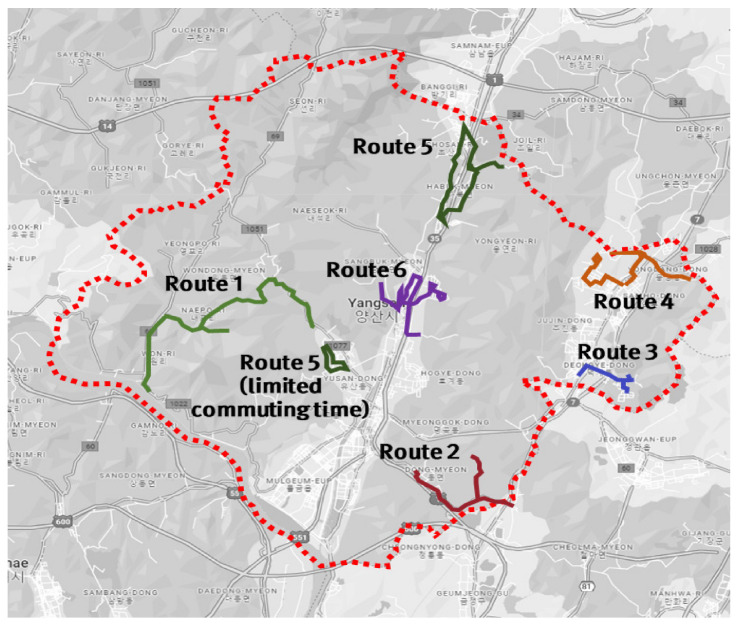
The distribution of the city-type bus routes in Yangsan-si.

**Figure 9 ijerph-19-11263-f009:**
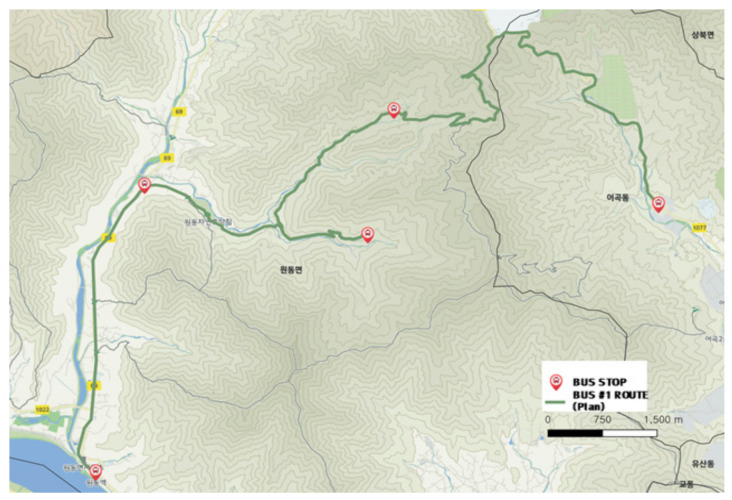
City-type bus No. 1 route.

**Figure 10 ijerph-19-11263-f010:**
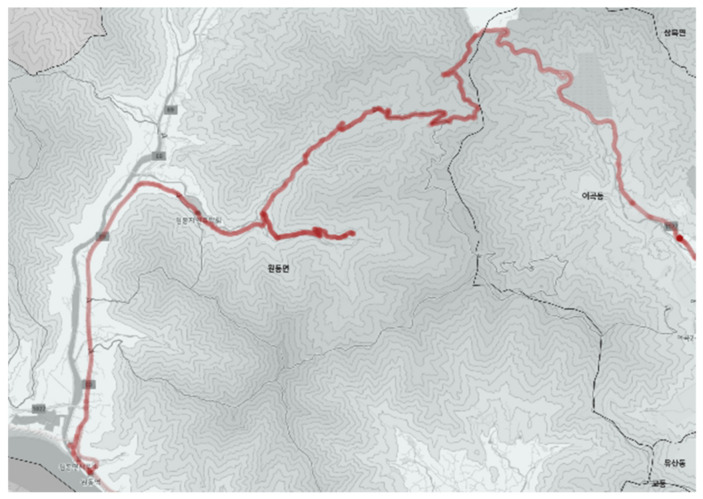
Actual driving record of city-type bus No. 1.

**Figure 11 ijerph-19-11263-f011:**
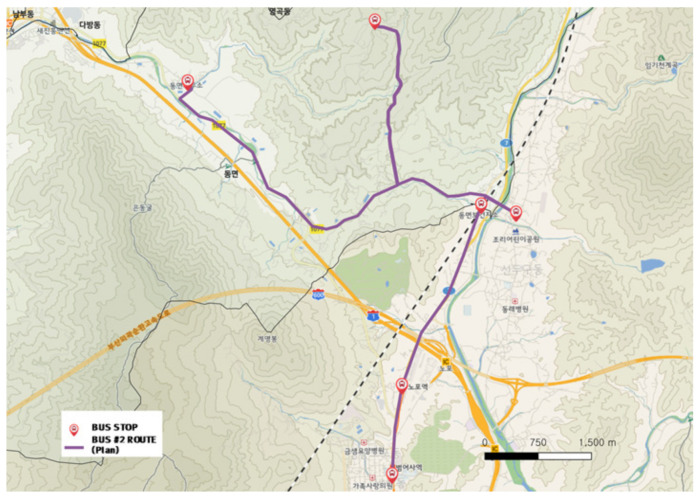
City-type bus No. 2 route.

**Figure 12 ijerph-19-11263-f012:**
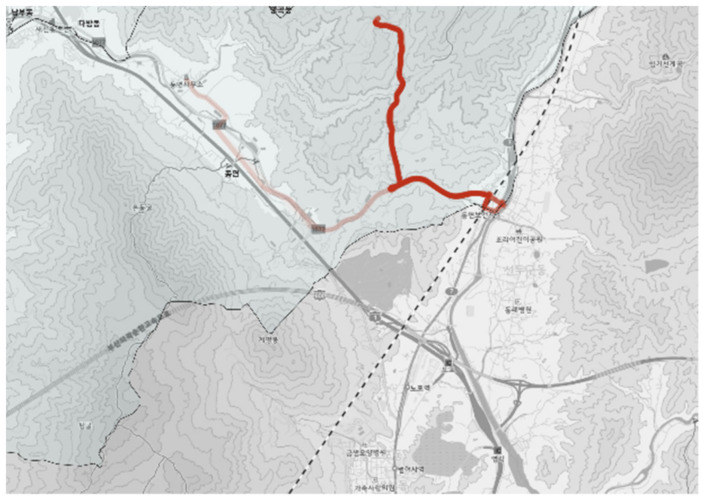
Actual driving record of city-type bus No. 2.

**Figure 13 ijerph-19-11263-f013:**
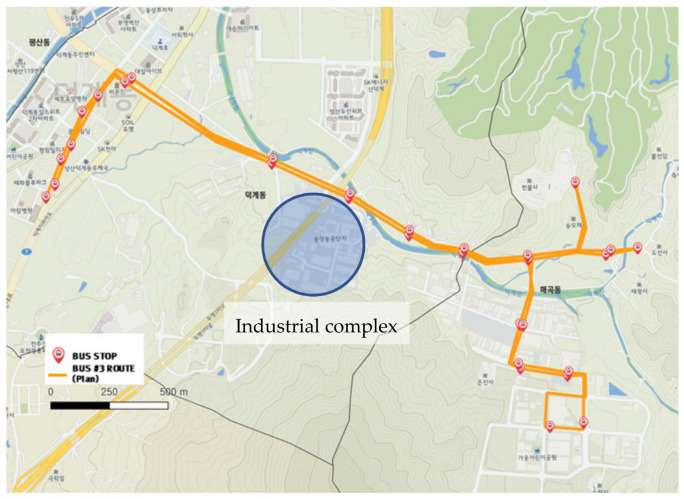
City-type bus No. 3 route (industrial area added).

**Figure 14 ijerph-19-11263-f014:**
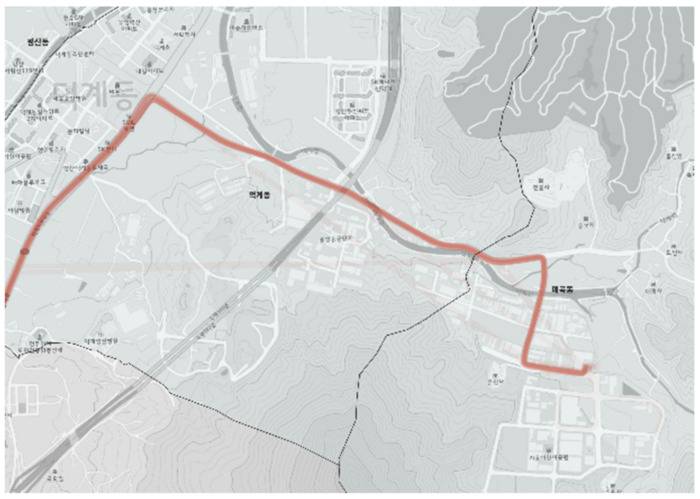
Actual driving record of city-type bus No. 3.

**Figure 15 ijerph-19-11263-f015:**
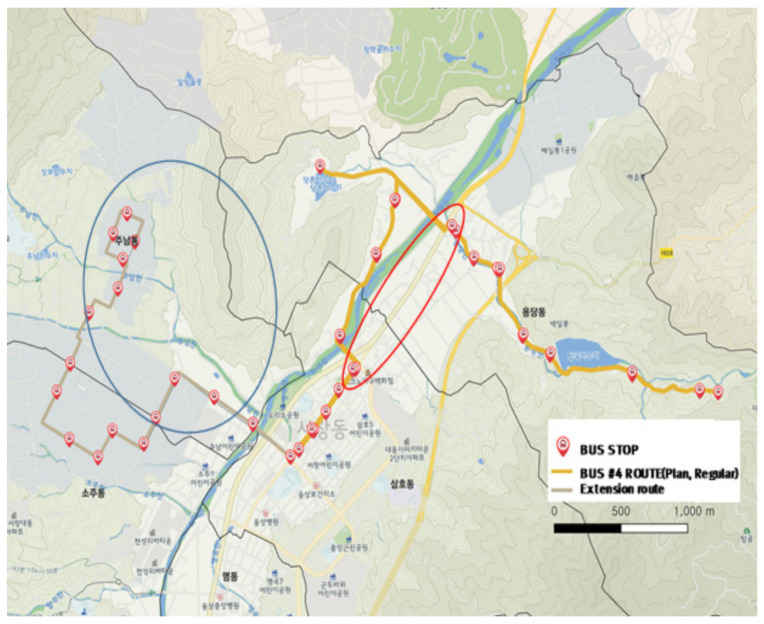
City-type bus No. 4 route.

**Figure 16 ijerph-19-11263-f016:**
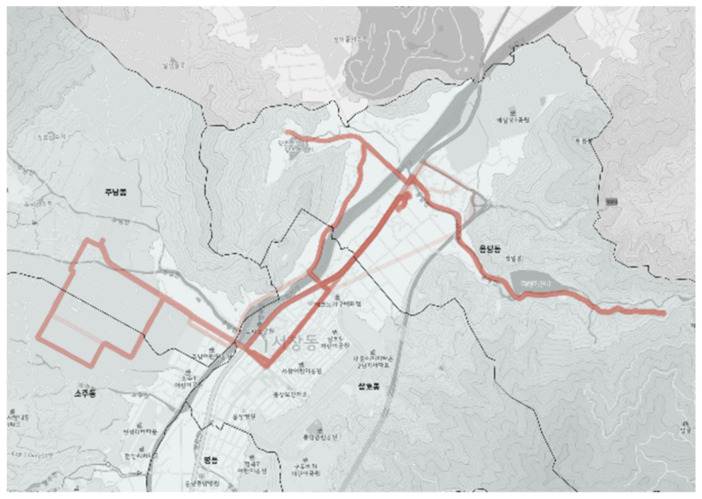
Actual driving record of city-type bus No. 4.

**Figure 17 ijerph-19-11263-f017:**
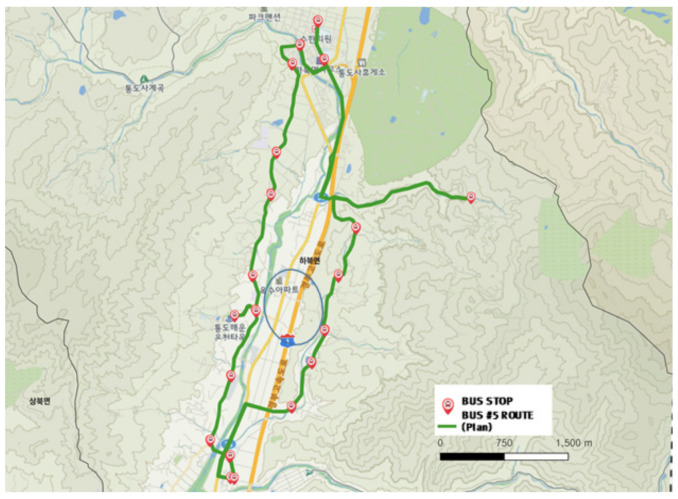
City-type bus No. 5 (general) route.

**Figure 18 ijerph-19-11263-f018:**
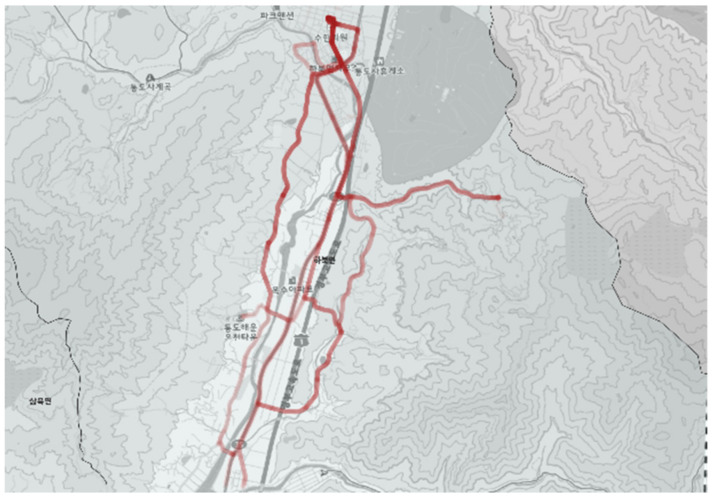
Actual driving record of city-type bus No. 5 (general).

**Figure 19 ijerph-19-11263-f019:**
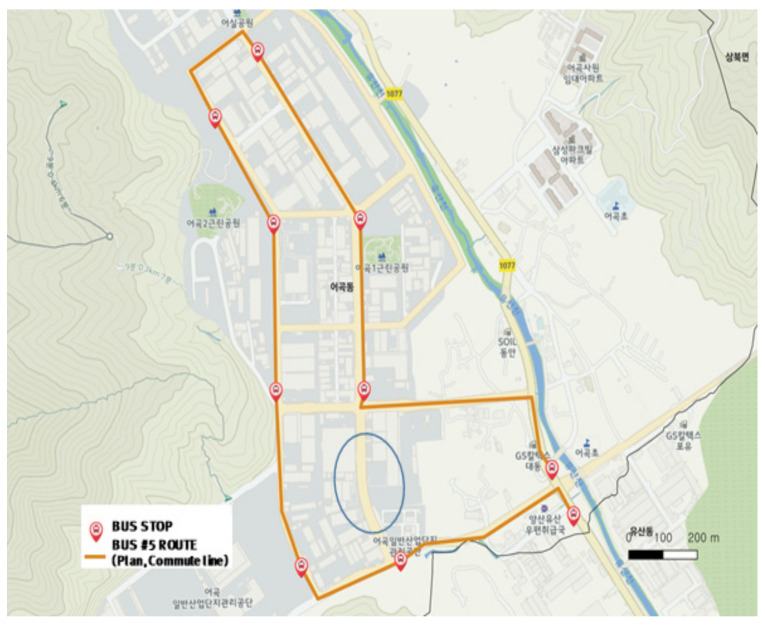
City-type bus No. 5 (commuting) route.

**Figure 20 ijerph-19-11263-f020:**
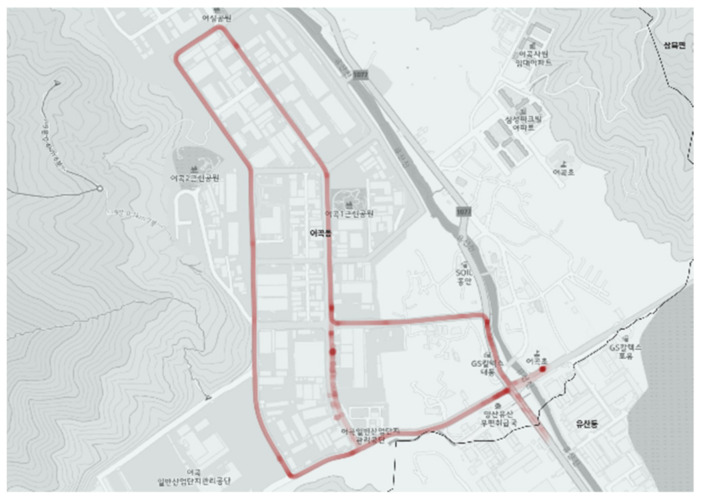
Actual driving record of city-type bus No. 5 (commuting).

**Figure 21 ijerph-19-11263-f021:**
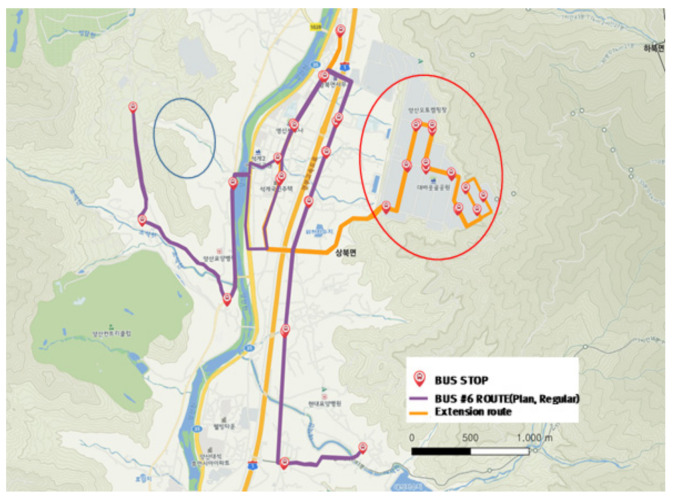
City-type bus No. 6 route.

**Figure 22 ijerph-19-11263-f022:**
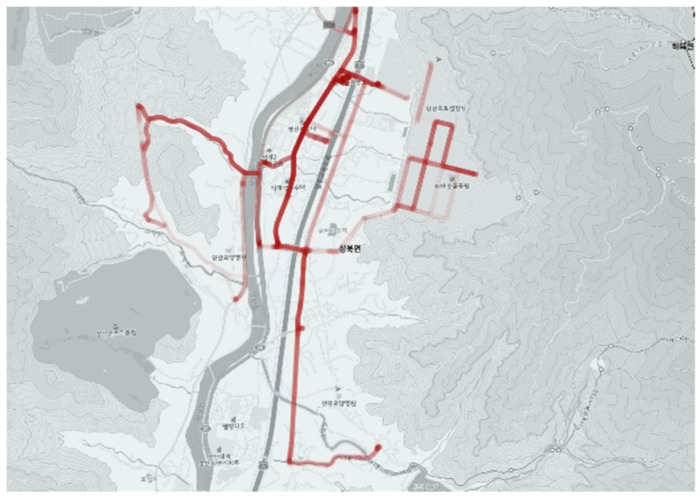
Actual driving record of city-type bus No. 6.

**Figure 23 ijerph-19-11263-f023:**
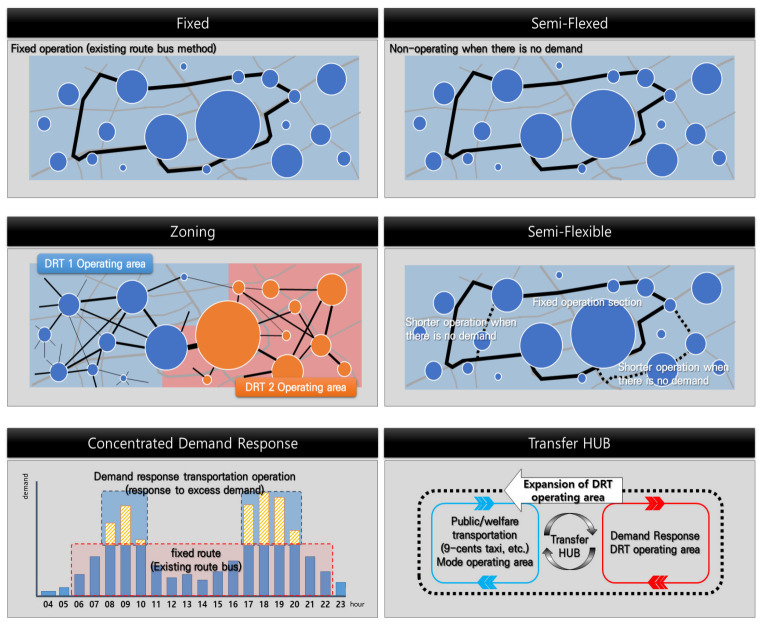
Methods of how to operate DRT means [32].

**Table 1 ijerph-19-11263-t001:** City-type bus’s operation methods and features by the route in Yangsan-si.

Route No.	Operation Method	Vehicle	Characteristics
1	Wednesday/Sunday: not in service; Lunar new year and Chuseok long holiday: not in service; lunchtime (12:00–13:00: not in service)	Seating capacity of 11 passengers	All-day on-demand method
2	Wednesday/Sunday: not in service; Lunar new year and Chuseok long holiday: not in service; lunchtime (12:00–13:00: not in service) (serviced twice per day)	Seating capacity of 15 passengers	Regular route full-time operation
3	Oesan and Maegok are fixed among the daytime routes (serviced eight times per day)	Seating capacity of 24 passengers	Some sections are DRT
4	Yongdang and Dangchon are fixed among the daytime routes (serviced seven times per day)	Seating capacity of 24 passengers	Some sections are DRT
5	None in service if no booking; lunchtime (11:40–12:40: not in service, no booking is made) (serviced eight times per day)	Seating capacity of 24 passengers	Not in service if no demand
6	None in service if no booking; lunchtime (11:40–12:40: not in service, no booking is made) (serviced seven times per day)	Seating capacity of 24 passengers	Not in service if no demand

**Table 2 ijerph-19-11263-t002:** Service frequency and the number of passengers by route and month (internal data of Yangsan-si).

Reference Year and Month	City-Type 1		City-Type 2		City-Type 3		City-Type 4		City-Type 5		City-Type 6	
	Service Frequency	No. of Passengers	Service Frequency	No. of Passengers	Service Frequency	No. of Passengers	Service Frequency	No. of Passengers	Service Frequency	No. of Passengers	Service Frequency	No. of Passengers
December 2020	107	60	80	185	71	95	38	37				
January 2021	87	36	83	167	203	331	95	67				
February	70	29	58	139	200	361	92	91				
March	129	75	85	288	244	476	118	100	545	2490	229	373
April	146	117	81	316	239	482	125	122	674	3522	298	481
May	158	111	79	282	234	532	141	176	671	3314	293	450
Jun.	136	72	85	265	237	526	170	216	652	2938	298	532
July	140	90	77	232	239	537	176	184	642	2919	302	609
August	113	48	73	204	220	433	179	193	642	2505	294	542
September	98	43	78	220	225	515	168	205	637	2969	286	687

**Table 3 ijerph-19-11263-t003:** DTG data collection information and unit [31].

Collection Information	Collection Unit
Model name	Product serial No.
Vehicle No.	(New)00GA0000/(Old)Seoul00GA0000
Vehicle identification No.	KL000000000000000 (17 digits)
Vehicle registration No.	Vehicle registration No. (12 digits)
Transport business registration No.	1234567890 (10 digits)
Driver license No.	1234567890
Driver employee No.	1234567
Driver name	ABC (initial in English)
Vehicle type	Intra-city bus (11)/Rural and fishing village bus (12)/Town bus (13)/Intercity bus (14)/Express bus (15)/Charted bus (16)/Special passenger bus (17)/General taxi (21)/Privately owned taxi (22)/General cargo (31)/Individual cargo (32)/Non-business purpose (41)
Daily mileage	km/day (Distance per day from 00:00 to 24:00, Range: 0000–9999)
Cumulative mileage	km (Cumulative distance from the first vehicle registration date, Range: 000000–999999)
Information generation time	YYYYMMDDhhmmsss (Year·Month·Day·hour·min·0.01 s)
Vehicle driving speed	km/h (Range: 000–255)
RPM (revolution per min)	Times/min
Brake signal	1 (on)/0 (off)
Vehicle location information	GPS X and Y coordinates (indicated by a decimal number, e.g., 127.123456 * 1,000,000 = 127123456)
Azimuth	Vehicle’s GPS azimuth (0–360)
Acceleration	m/s^2^
No. of sudden acceleration	Times (set the reference acceleration (km/h) and count the number of times above the reference acceleration)
No. of sudden breaking	Times (set the reference braking (km/h) and count the number of times above the reference braking)
No. of speeding	Times (set the reference speed (km/h) and count the number of times above the reference speed)
No. of events	Times
Fuel amount	L
Travel path	The path is indicated on a map
Equipment status	Tachograph normal (00)/Location tracking device GPS receiver abnormal (11)/Speed sensor abnormal (12)/RPM sensor abnormal (13)/Brake signal detection sensor abnormal (14)/Sensor input device abnormal (21)/Sensor output device abnormal (22)/Data output device abnormal (31)/Communication device abnormal (32)/Mileage calculation abnormal (41)/Power supply abnormal (99)

**Table 4 ijerph-19-11263-t004:** Example of DTG data.

Example of DTG Data
21092500000100|DT- 202,5.07.02|KMJTA18FPEC018515|11|6088102363|0000001|85|693058|6|602|0|128607091|35256553|34|70.0|70.9| 00|4812112900|21092509384500 221092500000100|DT- 202,5.07.02|KMJTA18FPEC018515|11|6088102363|0000001|85|693058|9|893|0|128607098|35256566|37|70.0|70.9| 00|4812112900|21092509384600 221092500000100|DT- 202,5.07.02|KMJTA18FPEC018515|11|6088102363|0000001|85|693058|12|1123|0|128607115|35256583|30|70.0|70.9| 00|4812112900|21092509384700 221092500000100|DT- 202,5.07.02|KMJTA18FPEC018515|11|6088102363|0000001|85|693058|14|1297|0|128607135|35256606|33|70.0|70.9| 00|4812112900|21092509384800

## Data Availability

Not applicable.

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
