# Peer review of "What Are More Efficient Transportation Services in a Rural Area? A Case Study in Yangsan City, South Korea"

_ijerph, 2022, doi:10.3390/ijerph191811263_

Round 1
Reviewer 1 Report
It's not very clear the focus on, let's say: 'geometry' of bus routes (see figures) without background information of population density spread, which may be significant for intensity of using intensity of specific parts of the routes.
Line 425: 'publication transport method'????
552: why 'We believe our study...' if there is the only one author?
Author Response
In order to check the characteristics and impact of the route, information on the area where the route is located has been supplemented. Added information for Figure 3 (regional location), Figure 4 (population density), and Figure 5 (senior distribution). The description and meaning of the picture are also presented.
Changes in words for typos and meaning transfer were implemented throughout the thesis.
Revisions are marked separately in the thesis (font color, etc.).
Thanks for the review.
Reviewer 2 Report
This paper presents six different bus dispatching services. Overall, the paper is well-motivated, and the background is pretty interesting. There are certain novelties of the proposed dispatching services. There are some serious issues of the paper:
- The writing quality of this paper should be seriously improved. I suggest the authors seek professional help if needed.
- The formatting of the paper is not consistent. Sometimes the texts are left justified, and the other time images are right justified. There are also some tables misplaced. Please be consistent and follow the formatting guidelines of the journal.
- The experimental sections need further clean-up. I also see no baselines, and the experimental metrics were not discussed properly. The authors should consider comparing their work with a multi-objective routing planning method as follows:
- Sarker, A., Shen, H. and Stankovic, J.A., 2018. MORP: Data-driven multi-objective route planning and optimization for electric vehicles. Proceedings of the ACM on Interactive, Mobile, Wearable and Ubiquitous Technologies, 1(4), pp.1-35.
- The conclusion could be shorter and to the point. Please, also add some discussion on future directions. Also, it would be interesting to see how the data-driven approaches shape the proposed approaches.
Author Response
Thanks for the review. Thank you for your kind comments, it was very helpful for further development of the thesis. Modifications and supplements have been implemented based on the comments below.
1. Overall inspection and writing of the thesis
A general inspection was performed including the form of the thesis. Based on the form provided by the journal, we checked the location of tables and figures, checked the method of writing, and implemented improvements. Duplicate parts of the content were checked and deleted, and additional content was included to convey meaning.
2. Analysis Related
A description of the spatial location of the area where the vehicle is being operated (Figure 3), the population distribution of the area (Figure 4), and the distribution of the elderly (Figure 5) have been added. The operation method for each route is different, and the contents and explanations for each route have been added accordingly. Since the routing method is demand response, it is a form of moving according to a pre-planned route when demand occurs. If degrees of freedom are granted here, we plan to apply the method of finding the shortest route in real time according to the regional characteristics (rural) where roads are narrow and alternative routes are limited. Because it is not a complex network, it is described based on the operation method rather than the routing part.
3. Complementary to the conclusion
Abbreviations and duplicates of the existing conclusions have been deleted. In addition, technology has been added for future research contents and directions.
For the parts that have been corrected and supplemented, the font color is presented differently in the text. Thank you
Round 2
Reviewer 2 Report
I am happy to see the currnet revision.